# Roflumilast prevented tissue damage caused by lipopolysaccharide-induced sepsis via anti-inflammatory action

Demet Yalçın Kehribar[1], Lale Saka Baraz[2]*, Seda Kırmızıkan[3], Emre Soner Tiryaki[4], Mustafa Nusret Çiçekli[5], Bahattin Avcı[6], Caner Günaydın[7]

1 Department of Internal Medicine, Dokuz Eylul University Faculty of Medicine, İzmir, Türkiye, 2 Department of Internal Medicine, Manisa City Hospital, Manisa, Türkiye, 3 Department of Histology and Embryology, Samsun University, Faculty of Medicine, Samsun, Türkiye, 4 Department of Physiology, Ondokuz Mayıs University, Faculty of Medicine, Samsun, Türkiye, 5 Department of Pharmacology, Samsun University, Faculty of Medicine, Samsun, Türkiye, 6 Department of Biochemistry, Ondokuz Mayıs University, Faculty of Medicine, Samsun, Türkiye, 7 Department of Genetic Medicine, Weill Cornell Medical College, New York, New York, United States of America

* lalesaka-6@hotmail.com

## Abstract

Sepsis is a life-threatening condition characterized by a dysregulated immune response leading to multiple organ dysfunction. Despite the use of antibiotics and anti-inflammatory drugs, recovery remains limited. Lipopolysaccharide (LPS), an endotoxin from Gram-negative bacteria, is widely used to mimic sepsis-like conditions in animals. This study investigated the anti-inflammatory and protective effects of Roflumilast at two doses (1.5 and 3 mg/kg) in a single-dose LPS-induced sepsis model. Sepsis was induced in rats by intraperitoneal injection of LPS (30 mg/kg), and Roflumilast was administered for 10 days. Liver and kidney injury were evaluated by serum alanine aminotransferase (ALT), aspartate aminotransferase (AST), blood urea nitrogen (BUN), and creatinine levels. Pro-inflammatory cytokines, including tumor necrosis factor (TNF), interleukin-1 beta (IL-1β), and interleukin-6 (IL-6), were measured using ELISA. Histopathological damage and kidney injury molecule-1 (KIM-1) expression were assessed in major organs. LPS significantly increased biochemical and cytokine markers, causing severe tissue damage. While 1.5 mg/kg Roflumilast showed no protective effects, 3 mg/kg markedly reduced inflammatory and injury markers, improved tissue architecture, and decreased KIM-1 expression. These findings suggest that a higher dose of Roflumilast effectively mitigates LPS-induced systemic inflammation and organ injury, supporting its potential as a therapeutic option for sepsis.

**Data availability statement:** All relevant data are within the manuscript.

**Funding:** The author(s) received no specific funding for this work.

**Competing interests:** The authors have declared that no competing interests exist.

## Introduction

Systemic inflammation and sepsis are major causes of morbidity and mortality throughout the globe. Secondary complications due to the inflammation systemically affect nearly all organs in the human body. Long-term exposure to inflammation is detrimental for chronic diseases such as asthma, diabetes, and chronic kidney disease [1]. Investigations mainly focus on the host's inflammatory response to find a way to overcome cell or organ damage [2]. Therefore, compounds that showed anti-inflammatory effects or regulated immune cells are candidates for possible novel treatment options. The use of inflammatory agents to mimic systemic inflammation is widely used to investigate novel mechanisms and novel anti-inflammatory agents in rodents [3]. Lipopolysaccharide is an endotoxin released from gram-negative bacteria and causes increased chemokine and cytokine expression in serum and tissue levels [4]. Because the response to LPS has been well documented, it has been extensively investigated in chronic and acute tissue injuries like sepsis.

Phosphodiesterases (PDEs) are a family of enzymes that catalyze the hydrolysis of 3',5'-cyclic adenosine monophosphate (cAMP) and 3',5'-cyclic guanosine monophosphate (cGMP) to their inactive 5'-AMP and 5'-GMP [5]. Phosphodiesterase-4 (PDE-4) inhibitors show broad anti-inflammatory effects and have been evaluated for use in treating different inflammatory diseases [6]. One of these inhibitors, roflumilast, is currently used to reduce the risk of chronic obstructive pulmonary disease (COPD) exacerbations in patients with severe COPD associated with chronic bronchitis and a history of exacerbations [7]. Although roflumilast has strong anti-inflammatory properties, its clinical use in humans may be limited by gastrointestinal adverse effects such as nausea and emesis. [8]. Additionally, knowledge about the effects of roflumilast on sepsis and organs other than the lungs remains limited. So, revealing the possible effects of roflumilast has the utmost importance in improving its therapeutic use and understanding its mechanism of action more deeply.

This study aims to investigate the potential effects of roflumilast on LPS-induced sepsis and to understand its effects on different organs. To achieve this goal, following LPS injection and roflumilast treatment, liver, lung, spleen, and kidney tissues were investigated after two different treatment doses of roflumilast.

## Materials and methods

### Animals

Thirty-two male Wistar albino (220–280) rats were used in this study. Animals were procured from Ondokuz Mayıs University Vivarium. Ethical approval was obtained from the Ethical Committee for Experimental Animals Ondokuz Mayıs University Faculty of Medicine (OMU HADYEK/2021/40). All experiments were carefully conducted according to the Guide for the Care and Use of Laboratory Animals adopted by the National Institutes of Health (USA) and reported as described in the ARRIVE guideline [9]. Animals were maintained in standard conditions (22 ± 0.5 °C, 55% humidity, 12/12 day and night cycle), 4–5 per cage till the end of the experiments. All animals were monitored for clinical signs of sepsis and distress throughout the experiment.

Environmental enrichment and minimal handling were provided to reduce stress. All procedures were carried out by trained personnel to minimize pain and distress.

Humane endpoints (rapid weight loss, severe lethargy, respiratory distress, inability to reach food/water, moribund state) were predefined, and animals were monitored at least twice daily (every 4 h during the first 24 h after LPS). Euthanasia was planned to be performed within 30 minutes of reaching endpoint criteria using ketamine/xylazine anesthesia followed by a secondary method, in accordance with the approved protocol. No animals reached endpoint criteria or died spontaneously; all animals were euthanized at the planned endpoint.

## Chemicals and experimental groups

Roflumilast, LPS, and dimethyl sulfoxide (DMSO) were purchased from Sigma Co. (Bethesda, US). Saline was obtained from a local pharmacy. LPS (*Escherichia coli*, serotype 0127: B8) was dissolved in the saline solution. Roflumilast was freshly dissolved just before the administrations in saline: DMSO (98:2, v:v). LPS and roflumilast treatment doses were selected according to the previous studies [10–11]. Animals were equally divided into four groups (Control, LPS, LPS+Roflumilast 1.5, LPS+Roflumilast 3). Sepsis was triggered by a single LPS (30 mg/kg, i.p.) injection. Roflumilast (1.5 and 3 mg/kg, p.o., 1 ml/kg) was administered with oral lavage one hour before a single LPS injection and continued for ten days, based on the pharmacokinetic knowledge in the literature [12]. Control animals were treated with saline (1 ml/kg, i.p.) as an LPS vehicle.

## Biochemical analysis

Twenty-four hours after the last treatments, animals were euthanized with anesthetized (Ketamine: xylazine, 80/10 mg/kg, i.p.) and transcardially perfused with heparinized saline after collecting blood samples from the tail veins. Animals were decapitated after perfusion. The tissues of the lungs, liver, kidneys, and spleen were carefully isolated. Serum samples were centrifuged ($300 \times g$) for five minutes to collect serum. Serum levels of creatinine (Cr), blood urea nitrogen (BUN), aspartate transaminase (AST), and alanine transaminase (ALT) were determined using a Cobas Mira Plus CC Chemistry Analyzer (Switzerland). TNF (E-EL-R0019, Elabscience, US), IL-1β (E-EL-R0012, Elabscience, USA), and IL-6 (E-EL-R0015, Elabscience, USA), levels in the serum samples were measured with commercially available ELISA kits, strictly following instructions.

## Histological and immunohistochemical analysis

Tissue samples were immediately fixed in 4% paraformaldehyde solution. Following 8-hour incubation, samples were washed under tap water, and a routine histology procedure was performed [13]. After tissues were embedded to paraffin blocks, five µm thick sections were stained with hematoxylin and eosin and examined under a light microscope for histopathological scoring. Lung, liver, and spleen scores were determined by randomly selecting 10 sections for each tissue sample. The score for each tissue sample represents the mean score of 10 different sections. The scoring system was as follows: none = 0, minimal = 1, mild = 2, significant = 3, severe = 4. The lung tissue was scored for alveolar wall thickness, hemorrhage, and inflammatory cell infiltration. The liver tissue was scored for degenerated hepatocytes, hemorrhage, and inflammatory cell infiltration. The spleen tissue was scored for the destruction of the architecture of red pulp, white pulp, and capsule, and for the destruction of the spleen tissue [14]. For immunohistochemical analysis, KIM-1 immunopositivity on five µm kidney sections was investigated by streptavidin-biotin-immunoperoxidase complex (Lab Vision™ UltraVision™ LP Detection System, Thermo Fisher Scientific) [15]. Samples were deparaffinized and dehydrated through graded alcohols, and then endogenous peroxidase activity was blocked with a 3% hydrogen peroxide solution. Next, sections were rinsed with phosphate-buffered saline (PBS, pH 7.2) and heated for 15 minutes in 95−98 °C citrate buffer, at 400 W for 7 min, then at 300 W for 5 min. for antigen retrieval. Sections were blocked by Ultra V block (Thermo Scientific, US)

and incubated with the anti-KIM1 antibody at 4°C overnight. Subsequent secondary antibody incubation was performed (Biotinylated goat anti-polyvalent) for 10 min at room temperature. Labeling was performed by 3-amino-9-ethyl carbazole (AEC) as the chromogen. Sections were counterstained with Mayer's hematoxylin and examined under a light microscope (Nikon Eclipse E400, Japan). KIM-1 immunopositivity in each section was evaluated under ×40 magnification fields per kidney sample. Immunohistochemistry was performed on a section from each tissue sample. KIM-1 immunopositive cell numbers were counted in 5 successive areas within each section at 40X magnification. KIM-1 immunoreactivity was quantified as the integrated density using ImageJ software (version 1.53).

### Statistical analysis

All experimental data were collected in SPSS (v21.0, Illinois, US). Data distribution was evaluated with Shapiro-Wilk's normality test. One-way ANOVA and Kruskal-Wallis were used to determine statistical differences. Tukey's and Mann-Whitney U tests were used for multiple comparisons. P values less than 0.05 were accepted as significant.

### Results

AST, ALT, BUN, and creatinine levels were measured to assess liver and kidney function after LPS injection. Our results demonstrated that LPS-injection caused a significant increase in serum levels of AST (196 ± 16.7, Fig 1A), ALT

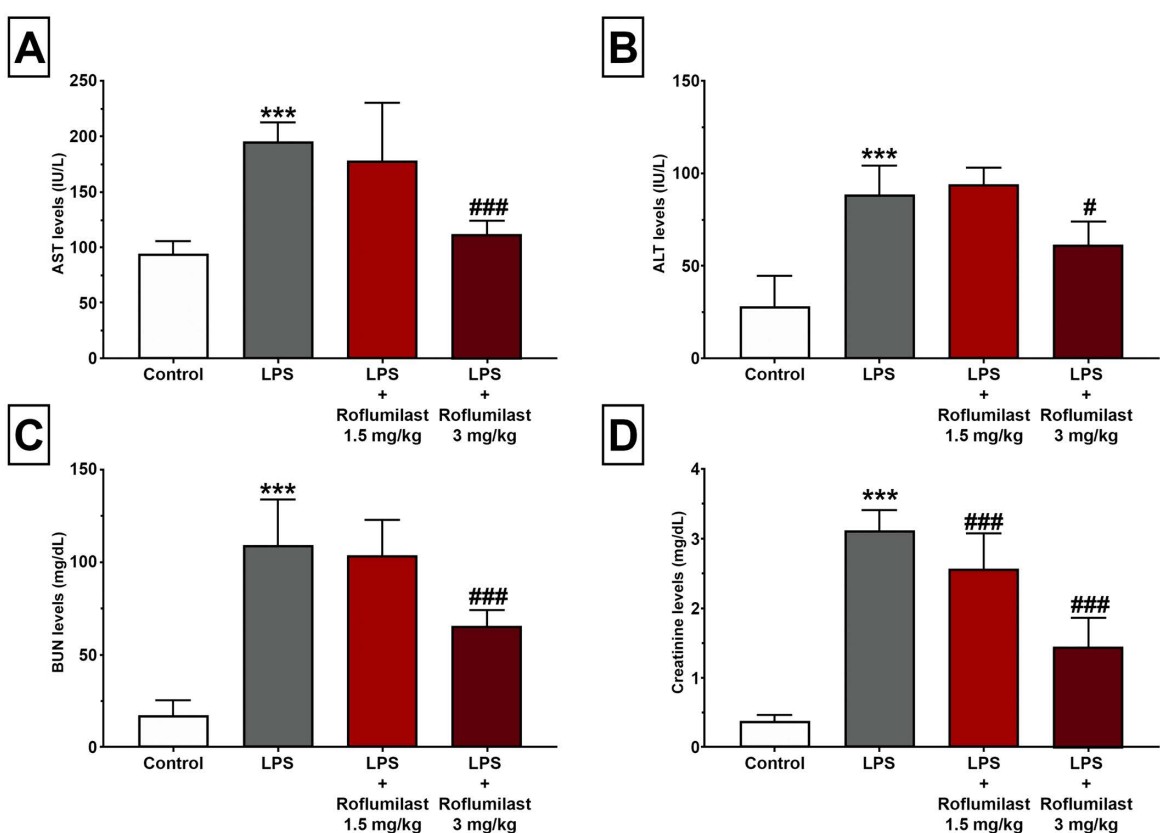

**Fig 1. Roflumilast inhibited the LPS-induced increase in biochemical liver and kidney markers.** AST **(A)**, ALT **(B)**, BUN **(C)**, and Creatinine **(D)** levels in all experimental groups. LPS caused a significant increase in all parameters. Although roflumilast 1.5 mg/kg failed to show any significant effect, roflumilast at the dose of mg/kg significantly prevented an LPS-induced increase in AST **(A)**, ALT **(B)**, BUN **(C)**, and creatinine **(D)**. All data expressed as mean±SD for eight animals per group. *** p < 0.001 versus control, ###p < 0.001 and #p < 0.05 versus LPS.

(88.3±15.9, Fig 1B), BUN (109.0±24.7, Fig 1C), and creatinine (3,11±0.3, Fig 1D) compared to control (94.3±10.9, 28.0±16.8, 17.3±8.0, 0.38±0.07, respectively, Fig 1) as expected. At the dose of 1.5 mg/kg, roflumilast treatment did not significantly affect AST (178±51.6, Fig 1A), ALT (94.0±9.52, Fig 1B), BUN (103.0±19.5, Fig 1C), and creatinine (2.58±0.4, Fig 1D) levels compared to LPS (p>0.05, Fig 1). But roflumilast, at the dose of 3 mg/kg, significantly inhibited LPS-induced increase in AST (112.0±11.7, Fig 1A) ALT (61.2±12.7, Fig 1B), BUN (65.7±8.51, Fig 1C), creatinine (1.46±0.4, Fig 1D) compared to control (p<0.001, p=0.015, p<0.001, p<0.001, respectively, Fig 1).

Serum levels of TNF, IL-1β, and IL-6 were determined by ELISA. LPS significantly increased TNF (111±19.2, Fig 2A), IL-1β (1099±236, Fig 2B) and IL-6 (55.0±4.16, Fig 2C) levels compared to control (18.6±5.37, 164±30.5, 16.8±4.26,

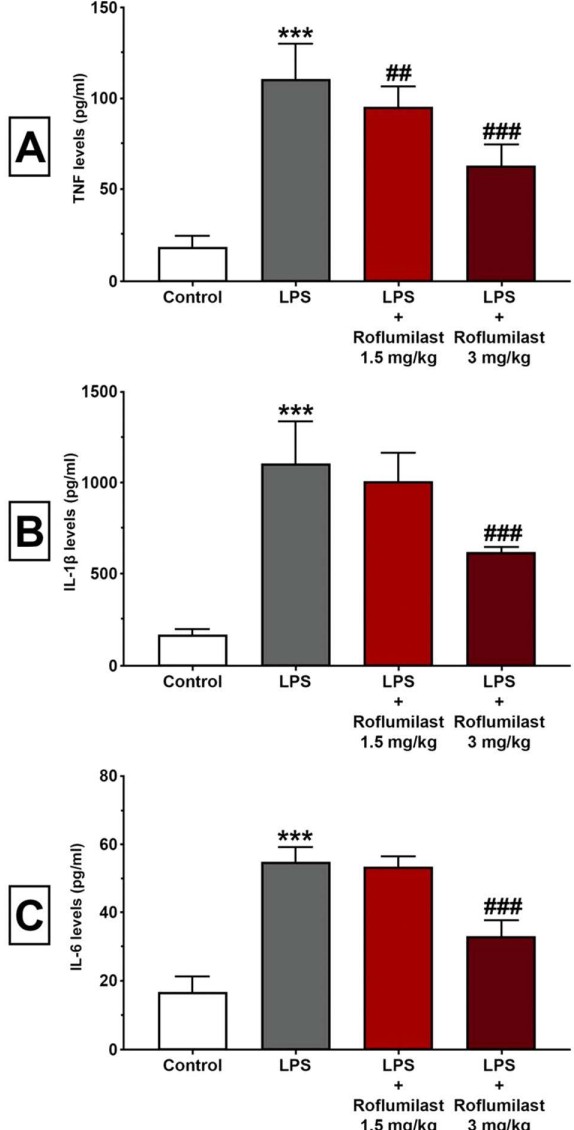

**Fig 2. Roflumilast prevented the LPS-induced increase in inflammatory cytokines.** TNF **(A)**, IL-16 **(B)**, and IL-6 (C) serum levels in all experimental groups. LPS caused a significant increase in all these pro-inflammatory cytokines. Although roflumilast 1.5 mg/kg failed to show any significant effect, roflumilast at the dose of 3 mg/kg significantly prevented an LPS-induced increase in TNF **(A)**, IL-16 **(B)**, IL-6 **(C)**. All data expressed as mean±SD for eight animals per group. *** p<0.001 versus control, ###p<0.001 and ##p<0.01 versus LPS.

p < 0.001, Fig 2). Roflumilast 1.5 mg/kg, like liver and kidney marker results, failed to show a significant effect against LPS-insult on TNF (95.2 ± 11.0, p > 0.05, Fig 2A), IL-1β (1010 ± 150, p > 0.05, Fig 2B) and IL-6 (53.3 ± 3.05, p > 0.05, Fig 2C). In contrast, roflumilast at the dose of 3 mg/kg significantly abolished the effect of LPS on TNF (63.6 ± 11.0, p < 0.001, Fig 2A), IL-1β (618 ± 26.5, p < 0.001, Fig 2B) and IL-6 levels (33.1 ± 4.72, p < 0.001, Fig 2C).

Histopathological examination was performed on lung, liver, spleen, and kidney tissues from all experimental groups. Our results demonstrated that LPS caused significant tissue damage and an increase in the histological scores in the lung (3.33 ± 0.18, Fig 3A, Fig 4, Table 1), liver (4.00 ± 0.21, Fig 3B, Fig 4, Table 1), and spleen (4.50 ± 0.19, Fig 3C, Fig 4, Table 1) tissues compared to control (0.50 ± 0.19, 0.00 ± 0.00, and 0.33 ± 0.18, respectively, p < 0.001, Fig 3, Fig 4, Table 1). Additionally, LPS significantly increased the number of KIM-1-positive tubules (45.30 ± 3.61, Fig 3D, Fig 5, Table 1) compared with control, indicating kidney damage. Although roflumilast at the dose of 1.5 mg/kg did not show any significant change (p > 0.05, Fig 3, Fig 4), roflumilast 3 mg/kg significantly prevented LPS-induced increases in histopathological scores in the lung (1.50 ± 0.19, p < 0.001, Fig 3A, Fig 4, Table 1), liver (2.50 ± 0.19, p < 0.001, Fig 3B, Fig 4, Table 1), spleen (2.00 ± 0.22, p < 0.001, Fig 3C, Fig 4, Table 1), and KIM-1 expression (21.50 ± 1.98, p < 0.001, Fig 3D, Fig 5, Table 1). Detailed histopathological scoring data are presented in Table 1.

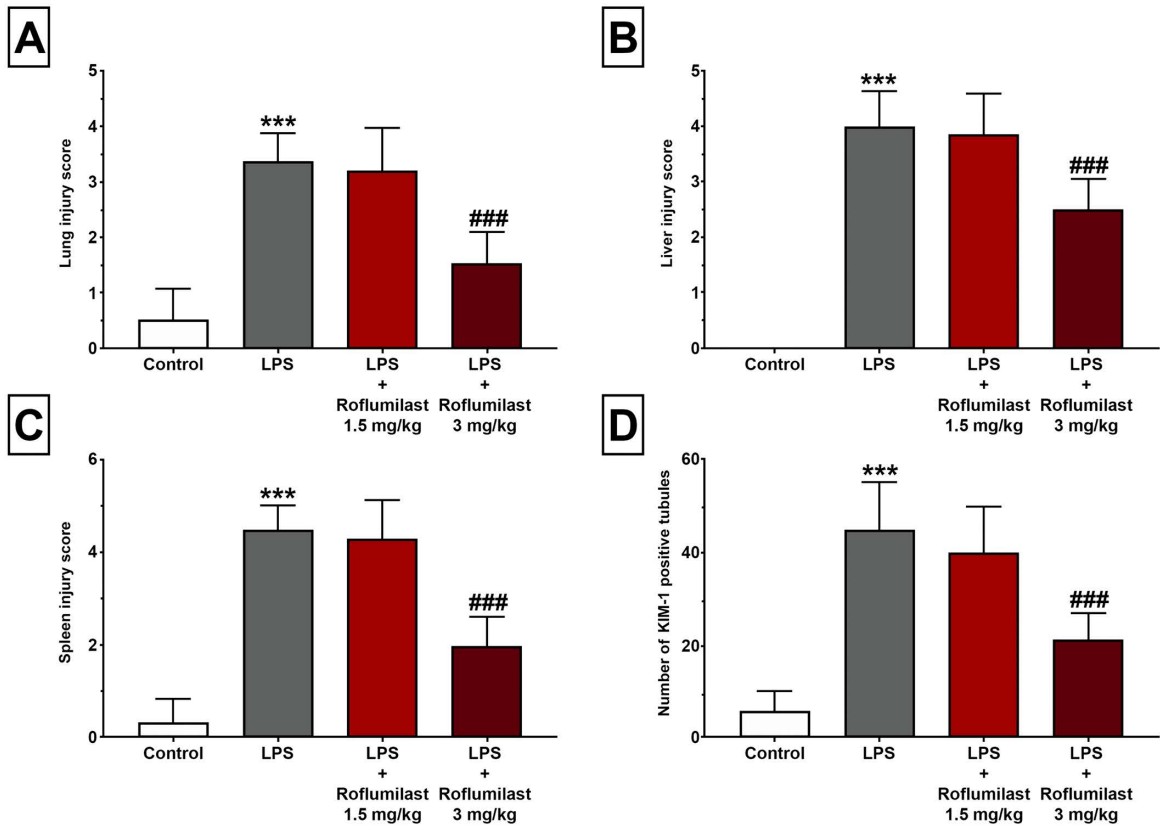

**Fig 3. Roflumilast ameliorated LPS-induced damage in the lung, liver, and spleen, and decreased KIM-1 expression in the kidney.** Lung **(A)**, liver **(B)**, spleen (C) histological scores, and KIM-1 positive tubules (D) in the kidney. LPS caused a significant increase in histological scores and KIM-1 immunopositivity. Although roflumilast 1.5 mg/kg failed to show any significant effect, roflumilast at the dose of 3 mg/kg significantly prevented an LPS-induced increase in histological scores and KIM-1 immunopositivity. All data expressed as mean ± SD for eight animals per group. *** p < 0.001 versus control, ###p < 0.001 and ##p < 0.01 versus LPS.

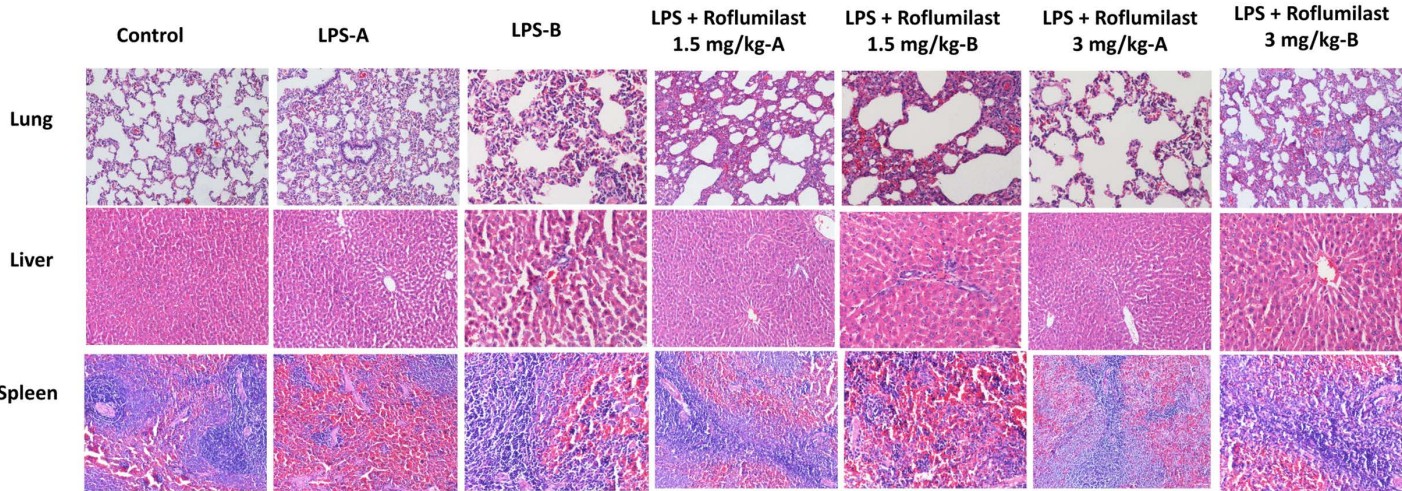

**Fig 4. Histological representation of the lung, liver, and spleen.** Histological representation of the lung, liver, and spleen. All tissues were stained with H&E. Substantial tissue damage was present in the LPS group. Roflumilast 3 mg/kg treatment alleviated this damage. Control and pictures A were taken under x10, and pictures B were taken under x20 magnification.

**Table 1. Histological scoring results (mean±SEM) of lung, liver, spleen, and kidney KIM-1 immunopositivity across experimental groups (n = 8).**

| Groups | Lung | Liver | Spleen | Kidney |
|---|---|---|---|---|
| Control | 0.50±0.19 | 0±0.0 | 0.33±0.18 | 5.83±1.62 |
| LPS | 3.33±0.18 | 4.00±0.21 | 4.50±0.19 | 45.30±3.61 |
| LPS+Roflumilast 3 mg/kg | 1.50±0.19 | 2.50±0.19 | 2.00±0.22 | 21.50±1.98 |

Data are presented as mean±SEM (n = 8). Histopathological damage was evaluated semi-quantitatively using a scoring system ranging from 0 to 4 (0 = none, 1 = minimal, 2 = mild, 3 = significant, and 4 = severe). Scores represent the average of ten randomly selected fields per tissue sample. Kidney injury was assessed by quantifying KIM-1 immunopositive tubules.

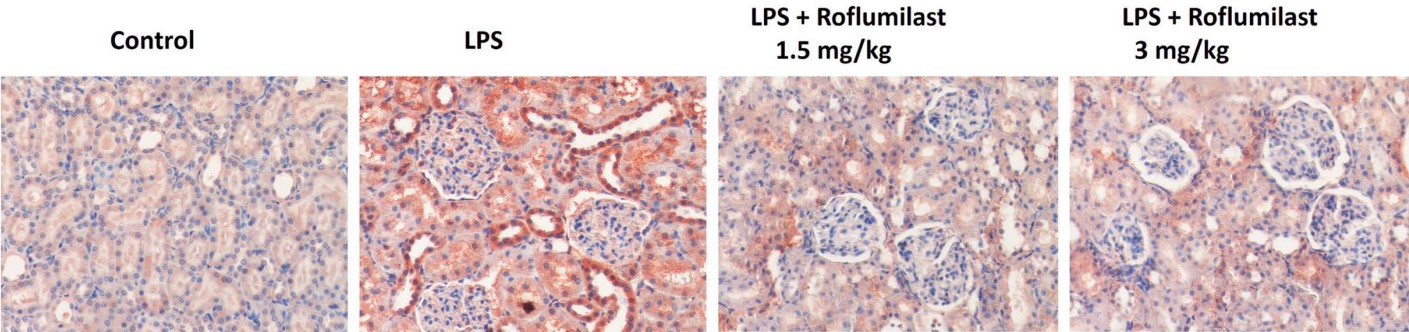

**Fig 5. KM-1 immunopositivity of the kidney.** KIM-1 immunopositivity was investigated in kidney tissue. Pictures were taken under x20 magnification.

## Discussion

In this study, the effect of roflumilast on LPS-induced sepsis was investigated. Roflumilast protected the lung, liver, spleen, and kidney against LPS-induced tissue damage. Roflumilast also prevented LPS-induced increases in biochemical

markers of liver and kidney damage. Similarly, roflumilast inhibited LPS-induced pro-inflammatory increase in serum. Our results suggest that roflumilast showed intense anti-inflammatory effects in LPS-induced sepsis.

PDE4 catalyzes the hydrolysis of cyclic nucleotides and regulates its downstream signaling. Inhibition of PDE4 with roflumilast or ropinirole results in inhibition of the degradation of cAMP and elevation of cAMP-response element-binding protein (CREB) phosphorylation [16]. This phosphorylation activates multiple cellular mechanisms and induces transcriptional activity in different cell types [17]. One of these transcriptional elements is NF-kB, which is widely known for regulating cellular response in the context of inflammation [18]. Therefore, inhibition of PDE4 with roflumilast has been shown to directly regulate cytokines via the NF-κB pathway [19]. LPS is recognized as an essential participant in the pathogenesis of sepsis and systemic inflammation [20]. LPS activates TLR4 and causes NF-kB p65 translocation to the nucleus to stimulate a pro-inflammatory response. Activated NF-kB in the nucleus promotes transcription of cytokines such as TNF-α, IL-6, and IL-1β [21]. These transcriptions are associated with increased levels of TNF-α, IL-6, and IL-1β, indicating ongoing inflammation. Our study demonstrated that a single LPS injection markedly increased serum cytokine levels, confirming ongoing systemic inflammation. This inflammatory condition possibly stimulates an immune response and leads to tissue damage and multiple organ failure. As well-known biomarkers of liver and kidney damage, we evaluated serum AST, ALT, urea, and creatinine levels. We found that LPS increased these markers. Our results showed that roflumilast (3 mg/kg) prevented increases in TNF-α, IL-6, and IL-1β, consistent with previous reports of its anti-inflammatory effects. Similarly, roflumilast (3 mg/kg) attenuated the increase in AST, ALT, urea, and creatinine levels, suggesting amelioration of liver and kidney damage. Although the dose used in this study is higher than the clinically recommended dose for COPD patients (500 μg/day) [7,16], higher weight-adjusted doses are commonly required in rodent models due to differences in metabolic rate and pharmacokinetics. Therefore, this dose is consistent with previous experimental studies demonstrating the anti-inflammatory and organ-protective effects of roflumilast in rat models [11,19].

Because sepsis affects multiple organs, we evaluated tissue histopathology of lungs, liver, spleen, and kidney tissues. Inflammation exaggerates cellular immune response owing to its origins in systemic infection [22]. That sustained systemic inflammatory response has been regarded as the primary mechanism behind the development of multiple organ failure [23]. Mainly, LPS-induced sepsis inflammation is related to acute lung, liver, spleen, and kidney damage [24]. Our study demonstrated that LPS caused histological damage in these tissues, in line with previous studies [25]. The protective effect of roflumilast on LPS-induced liver injury has been previously investigated, and our biochemical and histological results were similar to those reported in that study. Because roflumilast is currently used in COPD treatment, the protective effect of roflumilast has also been demonstrated in lung tissue [26]. Our study also verifies the protective effect of systemic roflumilast treatment on the lungs. However, as far as we know, this study is the first to investigate the possible anti-inflammatory effect of roflumilast treatment on inflammation in multiple organs. We also examined spleen and kidney tissues following drug treatment.

The spleen is a vital organ accepted as one of the central regulators of the immune response. Removing the spleen has been reported to lower the production of inflammatory cytokines and attenuate sepsis [27]. Additionally, under sepsis, damage in the spleen tissue due to the excessive stimulation and production of cytokines emerges. In light of this, we histopathologically examined spleen tissue and found that LPS caused significant damage. In line with lung and liver results, attenuation of pro-inflammatory cytokine levels with roflumilast treatment prevented spleen injury.

According to the current literature, the kidney is the most vulnerable organ in systemic inflammation, and sepsis-inflammation-associated acute inflammation may reach up to 70% [28]. Three primary changes are widely accepted as being responsible for this effect: inflammation, microcirculatory dysfunction, and metabolic cellular responses. In our study, increased BUN and creatinine levels indicate kidney damage, possibly related to elevated serum pro-inflammatory cytokine levels following LPS injection. We evaluated possible kidney damage with the recently described marker Kim-1, expressed on the proximal tubule apical membranes after injury. Our results demonstrated that LPS caused a significant increase in the KIM-1 positive tubules, which was also inhibited by roflumilast 3 mg/kg treatment.

The protective effects of PDE inhibitors in kidney injury have already been described [29]. Additionally, PDE4B has been specifically investigated, and PDE4B inhibitors have been shown to protect against inflammation-related kidney injury, which has a strong affinity for the PDE4B subtype [30].

Our study has two limitations. First, because NF-κB is the primary regulator of cytokine-mediated inflammatory damage, its role in the protective effect of roflumilast should be investigated at the organ- and tissue-specific level. Secondly, because cAMP directly interacts with CREB, the primary mechanism underlying the protective effect of roflumilast should be investigated for potential transcriptional changes.

## Conclusion

In conclusion, roflumilast treatment ameliorated sepsis and protected against inflammation-induced organ damage in the lung, liver, spleen, and kidney tissue. Our results suggest that this protection comes from the anti-inflammatory effects of roflumilast on pro-inflammatory cytokines. While roflumilast treatment appears to protect against sepsis, further studies are needed.

## Author contributions

**Conceptualization:** Demet Yalcin Kehribar, Lale Saka Baraz, Caner Günaydın.

**Data curation:** Demet Yalcin Kehribar, Lale Saka Baraz, Caner Günaydın.

**Formal analysis:** Demet Yalcin Kehribar, Caner Günaydın.

**Funding acquisition:** Demet Yalcin Kehribar, Bahattin Avci.

**Investigation:** Lale Saka Baraz, Seda Kirmizikan, Bahattin Avci.

**Methodology:** Lale Saka Baraz, Seda Kirmizikan, Bahattin Avci, Caner Günaydın.

**Project administration:** Seda Kirmizikan, Bahattin Avci, Caner Günaydın.

**Resources:** Seda Kirmizikan, Emre Soner Tiryaki, Bahattin Avci.

**Software:** Emre Soner Tiryaki, Mustafa Nusret Cicekli.

**Supervision:** Lale Saka Baraz, Emre Soner Tiryaki, Mustafa Nusret Cicekli.

**Validation:** Emre Soner Tiryaki, Mustafa Nusret Cicekli.

**Visualization:** Mustafa Nusret Cicekli.

**Writing – original draft:** Demet Yalcin Kehribar, Mustafa Nusret Cicekli, Caner Günaydın.

**Writing – review & editing:** Demet Yalcin Kehribar, Lale Saka Baraz.

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
