## [Decision Letter · Decision Letter 0]

22 Jan 2026

PONE-D-25-59084Roflumilast prevented tissue damage caused by lipopolysaccharide-induced sepsis via anti-inflammatory actionPLOS One

Dear Dr. Saka Baraz,

Thank you for submitting your manuscript to PLOS ONE. After careful consideration, we feel that it has merit but does not fully meet PLOS ONE’s publication criteria as it currently stands. Therefore, we invite you to submit a revised version of the manuscript that addresses the points raised during the review process.

We look forward to receiving your revised manuscript.

Kind regards,

Misbahuddin Rafeeq

Academic Editor

PLOS One

Journal Requirements:

2. Please remove your figures from within your manuscript file, leaving only the individual TIFF/EPS image files, uploaded separately. These will be automatically included in the reviewers’ PDF**.**

3. Please upload a copy of Supporting Information Figure 1, 2, 3, 5, which you refer to in your text on page 15.

4. Please remove all personal information, ensure that the data shared are in accordance with participant consent, and re-upload a fully anonymized data set.

Additional Editor Comments:

The authors are encouraged to refine, and make the figures more scientifically robust.

Reviewers' comments:

Reviewer's Responses to Questions

**Comments to the Author**

1. Is the manuscript technically sound, and do the data support the conclusions?

Reviewer #1: Partly

Reviewer #2: No

2. Has the statistical analysis been performed appropriately and rigorously?

Reviewer #1: No

Reviewer #2: Yes

3. Have the authors made all data underlying the findings in their manuscript fully available?

Reviewer #1: Yes

Reviewer #2: Yes

4. Is the manuscript presented in an intelligible fashion and written in standard English?

Reviewer #1: No

Reviewer #2: Yes

5. Review Comments to the Author

Reviewer #1: 1. Line 92 states that Roflumilast was administered p.o. It would be helpful for the authors to indicate how the Roflumilast dosing was controlled.

2. Line 115 does not indicate the temperature at which antigen retrieval was performed. It would be helpful for the authors to provide additional information.

3. Line 177 provides histological scores. It would be helpful for the authors to describe their calculation methodology in detail or provide a reference to it in the text.

4. The planned timeframe of the study is unclear from the manuscript. Were the animals euthanized immediately after the completion of the Roflumilast treatment course, or was the observation continued for some time?

5. In lines 61-62, the authors report that the side effects of Roflumilast are nausea and emesis. It would be useful for the authors to explain why rats, which are known to be physiologically incapable of emesis, were chosen as an animal model.

Reviewer #2: This study is the first which investigated the possible anti-inflammatory effect of roflumilast treatment on inflammation in multiple organs during LPS- induced sepsis in rats. Authors in their study showed that administration of roflumilast (3 mg/kg) prevented that elevation of proinflammatory cytokines TNF-α, IL-6, and IL-1β and attenuated the increase in AST, ALT, urea, and creatinine levels, suggesting amelioration of liver and kidney damage. Information extended knowledge about mechanism of action of this drug more deeply, what is originality of this study.

Manuscript is written in standard English and scientifically correct style and figures are prepared in good quality.

Experiments on animals were performed according to ARRIVE guidelines and number of animals (8) is sufficient for valid statistical analyses. Methodology is of some analyses is not described in sufficient details.

Authors showed images after H+E staining what is not very specific to show tissue damage. expressed as injury score. It is necessary describe the procedure for calculating injury score on tissue sections. How many sections were evaluated from each sample per group? One image from each tissue sample is not sufficient. In the case of analysing more sections/animal of data presentation (recommended), data must be shown as SD ± SEM (standard error of mean).

What type of damage (necrosis, apoptosis) was observed? Conclusions about the cytoprotective effect of the roflumilast on tissues such as the liver and spleen would have higher validity if damage in these organs was also assessed using a specific molecular marker, e.g. for determining apoptosis.

121: “KIM-1 immunopositivity in each section was evaluated under ×40 and x100 high-power magnification fields per brain sample.“

Can you explain why do you mention here brain tissue?

Line 164: Figure 3. Lung (A), liver (B), spleen (C) histological scores, and KIM-1 positive tubules (D) in all experimental groups. Line 169: Fig.4: KIM-1 immunopositivity was investigated in kidney tissue.

It is confusing legend and should be modified as KIM-1 immunoreactivity was monitored only in kidneys.

How many sections per group were examined to calculate positively stained tubules? What software was used to measure immunoreactivity?

Is the dose of 3mg/kg the within the range recommended for therapy of patients with COPD ?

6. PLOS authors have the option to publish the peer review history of their article (what does this mean?). If published, this will include your full peer review and any attached files.

Reviewer #1: **Yes:** Aleksey M. Nagornykh

Reviewer #2: No

---

## [Author Response · Author response to Decision Letter 1]

18 Feb 2026

Response to reviewers

We would like to sincerely thank the Editor and the Reviewers for their careful evaluation of our manuscript and for their constructive comments and valuable suggestions. We have carefully addressed all comments and revised the manuscript accordingly. The changes are highlighted in the revised version.

Reviewer #1:

1. Line 92 states that Roflumilast was administered p.o. It would be helpful for the authors to indicate how the Roflumilast dosing was controlled.

We are very grateful for the reviewer’s opinions and warnings. The corrections were shown below in Line 92.

Roflumilast (1.5 and 3 mg/kg, p.o. 1 ml/kg) was administered with oral lavage.

2. Line 115 does not indicate the temperature at which antigen retrieval was performed. It would be helpful for the authors to provide additional information.

We appreciate this observation. The antigen retrieval procedure has now been clarified in the Methods section (Line 121). For antigen retrieval, the slides were heated for 15 minutes in a 95-98 °C citrate buffer, at 400 W for 7 min, then at 300 W for 5 min.

3. Line 177 provides histological scores. It would be helpful for the authors to describe their calculation methodology in detail or provide a reference to it in the text.

We thank the reviewer for this suggestion. We have now expanded the description of the histological scoring method in the Methods section (Line 110-115). Lung, liver, and spleen scores were determined by randomly selecting 10 sections for each tissue sample. The score for each tissue sample represents the mean score of ten different sections. The scoring system was as follows: none=0, minimal= 1, mild= 2, significant= 3, severe =4. The lung tissue was scored for the thickness of the alveolar wall, hemorrhage, and infiltration of inflammatory cells. The liver tissue was scored for degenerated hepatocytes, hemorrhage, and infiltration of inflammatory cells. The spleen tissue was scored for the destruction of the architecture of red pulp, white pulp, and capsule, for the destruction of the spleen tissue.

4. The planned timeframe of the study is unclear from the manuscript. Were the animals euthanized immediately after the completion of the Roflumilast treatment course, or was the observation continued for some time?

We appreciate the reviewer’s comment. The experimental timeline has now been clarified in the Methods section (Line 97). Animals were euthanized twenty-four hours after the last roflumilast administration. No additional observation period was performed beyond this planned endpoint.

5. In lines 61-62, the authors report that the side effects of Roflumilast are nausea and emesis. It would be useful for the authors to explain why rats, which are known to be physiologically incapable of emesis, were chosen as an animal model.

We thank the reviewer for this insightful comment. Although rats are physiologically incapable of emesis, the aim of the present study was not to evaluate gastrointestinal adverse effects of roflumilast but rather to investigate its anti-inflammatory and organ-protective effects in an established LPS-induced sepsis model. Rats are widely used and well-validated models for studying systemic inflammation and sepsis-related organ injury. The reference to nausea and emesis in the Introduction reflects known clinical adverse effects in humans and was included to provide pharmacological background information. We have now clarified this point in the revised Introduction to avoid potential misunderstanding (Line 60-61)

Reviewer #2:

This study is the first which investigated the possible anti-inflammatory effect of roflumilast treatment on inflammation in multiple organs during LPS- induced sepsis in rats. Authors in their study showed that administration of roflumilast (3 mg/kg) prevented that elevation of proinflammatory cytokines TNF-α, IL-6, and IL-1β and attenuated the increase in AST, ALT, urea, and creatinine levels, suggesting amelioration of liver and kidney damage. Information extended knowledge about mechanism of action of this drug more deeply, what is originality of this study.

Manuscript is written in standard English and scientifically correct style and figures are prepared in good quality.

Experiments on animals were performed according to ARRIVE guidelines and number of animals (8) is sufficient for valid statistical analyses. Methodology is of some analyses is not described in sufficient details.

*The authors showed images after H+E staining what is not very specific to show tissue damage expressed as injury score. It is necessary describe the procedure for calculating injury score on tissue sections. How many sections were evaluated from each sample per group? One image from each tissue sample is not sufficient. In the case of analysing more sections/animal of data presentation (recommended), data must be shown as SD ± SEM (standard error of mean).

We thank the reviewer for this valuable suggestion. We have now clarified the histological evaluation procedure in the Methods section.

Lung, liver, and spleen scores were determined by randomly selecting 10 sections for each tissue sample. The score for each tissue sample represents the mean score of ten different sections. The scoring system was as follows: none=0, minimal= 1, mild= 2, significant= 3, severe =4. The lung tissue was scored for the thickness of the alveolar wall, hemorrhage, and infiltration of inflammatory cells. The liver tissue was scored for degenerated hepatocytes, hemorrhage, and infiltration of inflammatory cells. The spleen tissue was scored for the destruction of the architecture of red pulp, white pulp, and capsule, for the destruction of the spleen tissue (Line 110-115).

The images of H&E staining were revised to include novel images (Figures 4-5).

Data are now presented as mean ± SEM (Table 1)

*What type of damage (necrosis, apoptosis) was observed? Conclusions about the cytoprotective effect of the roflumilast on tissues such as the liver and spleen would have higher validity if damage in these organs was also assessed using a specific molecular marker, e.g. for determining apoptosis.

We thank the reviewer for this insightful comment. In the present study, tissue damage was primarily evaluated based on histopathological examination using H&E staining, which allowed assessment of overall tissue architecture, inflammatory infiltration, and structural alterations rather than distinguishing specific cell death pathways such as apoptosis or necrosis. We acknowledge that the use of specific molecular markers for apoptosis would further strengthen the mechanistic interpretation of the cytoprotective effects of roflumilast. However, such analyses were beyond the scope of the present study and are considered an important direction for future investigations.

*Line 121: KIM-1 immunopositivity in each section was evaluated under ×40 and x100 high-power magnification fields per brain sample.“

Can you explain why do you mention here brain tissue?

Thank you for your attention. The word "brain" was mistakenly written instead of "kidney." The necessary correction has been made (Line 126).

*Line 164: Figure 3. Lung (A), liver (B), spleen (C) histological scores, and KIM-1 positive tubules (D) in all experimental groups. Line 169: Fig.4: KIM-1 immunopositivity was investigated in kidney tissue.

It is confusing legend and should be modified as KIM-1 immunoreactivity was monitored only in kidneys.

Thank you for your attention. The necessary correction has been made (Line 172).

*How many sections per group were examined to calculate positively stained tubules?

Immunohistochemistry was performed on a section from each tissue sample. KIM-1 immunopositive cell numbers were counted in 5 successive areas within each section at 40X magnification (Line 127-128).

*What software was used to measure immunoreactivity?

KIM -1 immunoreactivity was quantified as the integrated density using ImageJ software (version 1.53) (Line 129).

*Is the dose of 3mg/kg the within the range recommended for therapy of patients with COPD ?

We thank the reviewer for this important translational question. The selected dose of roflumilast (3 mg/kg) was based on previously published experimental studies investigating its anti-inflammatory and organ-protective effects in rodent models. Using standard allometric scaling methods recommended by regulatory authorities, this dose corresponds to a human equivalent dose of approximately 0.486 mg/kg. It is well recognized that higher weight-adjusted doses are required in rodents compared to humans due to differences in metabolic rate, pharmacokinetics, and drug clearance. Moreover, the aim of the present study was to evaluate the pharmacological anti-inflammatory effects of roflumilast in an experimental sepsis model rather than to replicate the exact clinical therapeutic dose. Similar dosing regimens have been widely used in previous preclinical studies investigating the anti-inflammatory and tissue-protective effects of roflumilast. This clarification has been added to the Discussion section of the revised manuscript. (Line 233-236)

---

## [Decision Letter · Decision Letter 1]

18 May 2026

Roflumilast prevented tissue damage caused by lipopolysaccharide-induced sepsis via anti-inflammatory action

PONE-D-25-59084R1

Dear Dr. Lale Saka Baraz,

We’re pleased to inform you that your manuscript has been judged scientifically suitable for publication and will be formally accepted for publication once it meets all outstanding technical requirements.

Kind regards,

Misbahuddin Rafeeq

Academic Editor

PLOS One

Additional Editor Comments (optional):

Reviewers' comments:

Reviewer's Responses to Questions

**Comments to the Author**

1. If the authors have adequately addressed your comments raised in a previous round of review and you feel that this manuscript is now acceptable for publication, you may indicate that here to bypass the “Comments to the Author” section, enter your conflict of interest statement in the “Confidential to Editor” section, and submit your "Accept" recommendation.

Reviewer #1: All comments have been addressed

2. Is the manuscript technically sound, and do the data support the conclusions?

Reviewer #1: Yes

3. Has the statistical analysis been performed appropriately and rigorously?

Reviewer #1: Yes

4. Have the authors made all data underlying the findings in their manuscript fully available?

Reviewer #1: Yes

5. Is the manuscript presented in an intelligible fashion and written in standard English?

Reviewer #1: Yes

6. Review Comments to the Author

Reviewer #1: Using a classic in vivo sepsis model, the authors clearly and thoroughly demonstrated the anti-inflammatory effect of Roflumilast. The dose-dependent effect was confirmed through biochemical and immunological studies of blood serum. The resulting multiple organ dysfunction and therapeutic effect were confirmed by histopathological and immunohistochemical methods in lung, liver, spleen, and kidney sections. The manuscript is recommended for publication, as the information presented by the authors may lead to new trends in sepsis treatment.

7. PLOS authors have the option to publish the peer review history of their article (what does this mean?). If published, this will include your full peer review and any attached files.

Reviewer #1: **Yes:** Aleksey M. Nagornykh

---

## [Editor Report · Acceptance letter]

PONE-D-25-59084R1

PLOS One

Dear Dr. Saka Baraz,

I'm pleased to inform you that your manuscript has been deemed suitable for publication in PLOS One. Congratulations! Your manuscript is now being handed over to our production team.

Kind regards,

on behalf of

Dr. Misbahuddin Rafeeq

Academic Editor

PLOS One